# Influence of Biomarkers on Mortality among Patients with Hepatic Metastasis of Colorectal Cancer Treated with FOLFOX/CAPOX and FOLFIRI/CAPIRI, Including Anti-EGFR and Anti-VEGF Therapies

**DOI:** 10.3390/medicina60061003

**Published:** 2024-06-19

**Authors:** Dorel Popovici, Cristian Stanisav, Laurentiu V. Sima, Alina Negru, Sergiu Ioan Murg, Adrian Carabineanu

**Affiliations:** 1Department of Oncology, Faculty of Medicine, Victor Babeş University of Medicine and Pharmacy Timisoara, Eftimie Murgu Square 2, 300041 Timisoara, Romania; dorel.popovici@umft.ro; 2Departments of Radiology, Faculty of Medicine, Victor Babeş University of Medicine and Pharmacy Timisoara, 300041 Timisoara, Romania; 3Department of Surgical Semiology, Faculty of Medicine, Victor Babeş University of Medicine and Pharmacy Timisoara, Eftimie Murgu Square 2, 300041 Timisoara, Romania; carabineanu.adrian@umft.ro; 4Department of Cardiology, Faculty of Medicine, Victor Babeş University of Medicine and Pharmacy Timisoara, Eftimie Murgu Square 2, 300041 Timisoara, Romania; 5Doctoral School of Medicine, Faculty of Medicine and Pharmacy, University of Oradea, 1 Decembrie Square 10, 410073 Oradea, Romania

**Keywords:** colorectal cancer, Body Mass Index (BMI), patient demographics, biomarkers, lymphocytes, NHL score, NLR, PLR, SII, survival analysis, ROC curve

## Abstract

*Background and objectives:* Colorectal cancer is a major global health concern, with a significant increase in morbidity and mortality rates associated with metastatic stages. This study investigates the prognostic significance of various clinical and laboratory parameters in patients with metastatic CRC. *Materials and Methods:* A retrospective cohort of 188 CRC patients with hepatic metastasis from the OncoHelp Association in Timisoara was analyzed from January 2016 to March 2023. Data on demographics, clinical characteristics, and biomarkers, such as lymphocyte counts, as well as various inflammation indices, were examined. Statistical analyses included univariate and multivariate logistic regression, Kaplan-Meier survival analysis, and ROC curve assessments. *Results:* Our findings indicate significant associations between survival outcomes and several biomarkers. Higher BMI and lymphocyte counts were linked with better survival rates, while higher values of Neutrophil-Hemoglobin-Lymphocyte (NHL) score, Neutrophil-Lymphocyte Ratio (NLR), Platelet-Lymphocyte Ratio (PLR), and Systemic Immune-Inflammation Index (SII) were predictors of poorer outcomes. Notably, the presence of hepatic metastasis at diagnosis was a critical factor, significantly reducing overall survival. *Conclusions:* The study has expanded the current understanding of prognostic factors in CRC, advocating for a multi-dimensional approach to prognostic evaluations. This approach should consider not only the traditional metrics such as tumor stage and histological grading but also incorporate a broader spectrum of biomarkers. Future studies should aim to validate these findings and explore the integration of these biomarkers into routine clinical practice, enhancing the precision of prognostic assessments and ultimately guiding more personalized treatment strategies for CRC patients.

## 1. Introduction

The battle against cancer remains one of the topmost challenges in global health, manifesting in diverse forms and affecting millions worldwide. In terms of incidence, breast cancer leads as the most common, followed by prostate cancer, with colorectal cancer ranking third. Despite its position in incidence, colorectal cancer notably ascends to the second place in terms of mortality, underscoring its severe impact on both men and women globally. In the year 2022, Europe observed 524,043 new instances of colorectal cancer (CRC), marking an age-standardized incidence rate of 29.5 per 100,000 people and a 3.5% likelihood of diagnosis by age 74. Such statistics situate Europe as a leading region for CRC occurrence, with only North America and Oceania showing higher age-standardized rates. The severity of CRC is further highlighted by the 243,488 lives it claimed in Europe within the same timeframe, revealing an age-standardized mortality rate of 11.9 per 100,000 and a 1.3% risk by age 74. Romania’s situation vividly depicts the problem that many European countries face in addressing CRC. The country reported 13,280 CRC cases in 2022, with an age-standardized rate of 31.5 per 100,000 and a raw incidence rate of 69.8 per 100,000, signifying a considerable health burden and positioning Romania above several European counterparts [1].

Looking ahead to 2022–2045, the anticipated rise in CRC cases by 70.5% worldwide, primarily due to population aging and growth, raises alarms. In Romania, projections indicate a 17.2% increase in CRC cases, escalating from 13,541 in 2022 to 15,871 by 2045, despite an expected demographic decline. This forecast stresses the urgent necessity for stronger prevention, earlier detection, and more effective treatments to curb the growing impact of CRC in Romania and throughout Europe [2].

### 1.1. Metastasis in Colorectal Cancer

The invasion-metastasis cascade and the role of metastasis-initiating cells (MICs) are critical elements in understanding the complex processes underlying CRC progression and dissemination. MICs represent a subpopulation within the tumor that possesses both stem-like qualities and invasive capabilities, making them pivotal in the metastatic process. These cells can self-renew and produce aberrant successors while co-opting the tumor microenvironment to support their survival and proliferation. The invasive nature of MICs is attributed to the activation of signaling pathways such as NOTCH, Wnt/β-catenin, and TGF-β. For instance, interactions between JAGGED on endothelial cells and NOTCH on tumor cells support tumor cell survival near blood vessels. Moreover, MICs exhibit resistance to TGF-β‘s cytostatic effects, possibly by inhibiting cytotoxic T cells and attracting immunosuppressive cells like myeloid-derived suppressor cells and regulatory T-cells [3]. The invasion-metastasis cascade comprises a series of steps that CRC cells undergo, beginning with detachment from the primary tumor site, facilitated by reduced expression of epithelial junction components like E-cadherin. These cells then invade the extracellular matrix (ECM) by secreting or activating enzymes such as matrix metalloproteinases and urokinase plasminogen activators, enabling their migration through tissue barriers. The degradation of pericellular and ECM structures is followed by the release of pro-angiogenic factors like VEGF, fostering the development of new blood and lymphatic vessels. This neovascularization aids tumor cells in intravasation into the circulation, from where they can travel to distant sites. During transit, tumor cells form clusters with other cells, such as leukocytes and platelets, enhancing their survival against shear stress and avoiding apoptosis induced by detachment from the ECM, a phenomenon known as anoikis. Upon reaching a suitable secondary site, these circulating tumor cells exit the bloodstream, seed into the new tissue, and enter a dormant state characterized by halted cell division and activation of survival pathways. Reactivation from this dormant state allows for outgrowth and colonization of secondary sites, mirroring processes observed in adult stem cells [3].

Liver metastasis significantly impacts CRC prognosis, driven by intricate molecular and cellular interactions. Research highlights the importance of the β2 subunit of LFA-1 integrin in modulating the liver microenvironment, suggesting a less tumor-promoting environment when its expression is reduced. Benedicto et al. explored the intricate role of β2 (CD18) integrin in the metastatic progression of colorectal cancer to the liver, utilizing a genetically modified murine colon carcinoma C26 cell line, dubbed β2-C26, characterized by reduced β2 integrin expression. Their findings highlight the critical function of the β2 subunit in LFA-1 (CD11a/CD18) integrin activity, which is instrumental in metastasis formation. The research reveals that diminished β2 integrin expression leads to a notable decrease in both the metastatic potential and the size of tumor foci in vivo when these modified cells are introduced intrasplenically into mice. The adhesion of C26 cells to liver sinusoidal endothelial cells is specifically impaired by the partial deficiency of β2 integrin or the neutralization of CD11a but not affected by blocking other adhesion molecules such as CD11b/c or VCAM-1 on LSECs. This underscores the pivotal role of LFA-1 in mediating the adhesive interactions critical for the establishment and progression of colorectal cancer metastases in the liver [4].

Also, the presence of cancer stem cell markers, notably CD133, CD44, and β-catenin, is notably linked to liver metastasis, underscoring a unique pathobiological pathway distinct from other metastatic sites like the peritoneum. Additionally, the expression of pluripotency factor Oct4 and the loss of Smad4 in a significant subset of metastatic CRC cases further delineate the aggressive nature of CRC cells prone to liver metastasis. The liver’s microenvironment, shaped by interactions with molecules such as ADAM9 secreted by hepatic stellate cells and the involvement of miRNAs like has-miR-31-5p, play crucial roles in facilitating CRC metastasis. These elements promote carcinoma invasion and epithelial-mesenchymal transition, highlighting the dynamic crosstalk between tumor cells and their surroundings [3,5].

Furthermore, cytokines like IL33, by recruiting myeloid cells that aid angiogenesis, indirectly support liver metastasis, emphasizing the bidirectional interactions between tumor cells and the microenvironment. Angiogenesis is pivotal, not only for the initial tumor cell dissemination but also for the establishment and growth of metastases in the liver, with the primary tumor playing a role in priming the liver for metastatic colonization [5].

### 1.2. Risk Factors and Screening Guidelines

A pivotal aspect of CRC diagnostics lies in the identification of risk factors, which are broadly categorized into lifestyle/behavioral and genetic. Age emerges as a significant, non-modifiable risk factor, with a substantial proportion of CRC cases occurring in individuals over 65 years of age. Nevertheless, there is a growing concern over the rising incidence in younger age groups, particularly those between 40 and 44 years, underscoring the need for vigilant screening protocols. High-risk individuals, including those with a personal or family history of adenomas, CRC, inflammatory bowel diseases, or those carrying inherited cancer syndromes, necessitate active screening and, in cases of genetic predispositions, referral for genetic counseling as per ESMO guidelines [6].

Screening for CRC employs a variety of modalities, ranging from stool-based tests to visual examinations of the colon and rectum. Stool-based tests, such as the fecal immunochemical test and the guaiac-based fecal occult blood test, offer non-invasive options, while structural exams like colonoscopy and CT colonography provide a more direct assessment of colorectal anatomy. The choice of screening test is influenced by factors including the individual’s risk category, personal preferences, and the specific recommendations of healthcare providers. It’s imperative that any abnormal findings from non-colonoscopy screening methods be promptly followed up with a colonoscopy to ensure accurate diagnosis. Individuals at an increased or high risk of CRC may require a more aggressive screening regimen, which could include starting screenings before the age of 45, undergoing screenings more frequently, or employing specific tests tailored to their risk profile [7].

### 1.3. Chemotherapy and Targeted Therapies in Metastatic CRC

Addressing the intricate landscape of metastatic colorectal cancer treatment, ASCO guidelines point to a multifaceted and personalized strategy. Central to this approach is the adoption of chemotherapeutic regimens, notably the doublet combinations of folinic acid, fluorouracil, and either oxaliplatin (FOLFOX) or irinotecan (FOLFIRI), and/or the capecitabine and either oxaliplatin (CAPOX) or irinotecan (CAPIRI), enriched with the integration of bevacizumab for specific unresectable microsatelite stable or proficient mismatch repair metastatic CRC cases. Jointly, the landscape is further refined by targeted therapies, such as pembrolizumab for distinct molecular profiles and anti-EGFR agents for left-sided RAS wild-type metastatic CRC, demonstrating a commitment to precision medicine. The guidelines also regard surgical and radiation interventions for select metastatic scenarios, emphasizing their judicious application. This comprehensive framework is anchored in shared decision-making, ensuring that therapeutic avenues resonate with patient preferences and the nuanced aspects of their condition [8].

However, ESMO guidelines introduce a prudent lens on these treatments, advocating for their cautious administration due to potential complications. This caution extends to the pre-treatment evaluation, notably Dihydropirimidine dehydrogenase testing to reduce the risk of severe toxicity from fluoropyrimidine-based chemotherapy, reflecting a coordinated move towards individualized care. The guidelines also deliberate on the complex interplay of age in treatment choices, particularly among the elderly, advocating for a prudent assessment of benefits versus risks [6]. Biomarkers are useful and should be included in the diagnostic plan as they help prognosis and can lead to a reduction of disease recurrence through the correct use of chemotherapies. Among the most studied biomarkers in CRC are MSI-H/dMMR, MAPK-pathway (Ras-Raf-MEK-ERK), HER2, APK, CEA, NTRK, DPD, ctDNA, PD-L1 pathway, WNT-pathway, RET, VEGF, and CTC. Recent studies also investigate serum biomarkers, such as neutrophils, lymphocytes, monocytes, platelets, hemoglobin, NLR, PLR, etc., for the prediction of chemotherapy cluster symptoms and disease progression [9,10]. The incorporation of personalized medicine, through biomarker and genetic profiling, into the treatment algorithm underscores a paradigm shift towards a more nuanced and patient-centric approach in metastatic CRC management.

### 1.4. Biomarkers

Inflammation plays a very important role in most diseases, especially cancer, which makes certain biological values to help measure the level of inflammation in these patients. In cancer, these biomarkers allow risk stratification in diagnosis and provide insight into the choice of therapy. Recent studies show that blood count changes occur months before diagnosis in patients with metastatic CRC compared to those without metastases [11]. Among the most commonly used biomarkers are hemoglobin, lymphocytes, neutrophils, and platelets.

Haemoglobin is mainly used to determine whether the patient is suffering from anemia, which can be multifactorial. Iron deficiency anemia is most common in diagnostic patients with CRC, which is caused by repeated bleeding due to tumor ulceration, low iron stores, and decreased transferrin. An important factor exacerbating this iron deficiency is malnutrition. Another way in which anemia occurs is through functional iron deficiency, which is mediated by increased levels of the inflammatory cytokines IL-1, -6, -8, and TNF-alpha, which cause increased hepcidin levels. Hepcidin inhibits intestinal absorption of iron and sequesters the remaining iron molecules in macrophages and hepatocytes. Lymphocytes and neutrophils are part of the innate immune response and are responsible for tumor-promoting and immune suppression in cancer patients. They are used to clinically observe the level of inflammation and have been associated with poor prognosis in patients diagnosed with cancer. The Neutrophil-to-Lymphocyte Ratio, Neutrophil-Hemoglobin-Lymphocyte score, Platelet-to-Lymphocyte Ratio, and systemic inflammatory index are indicators that reflect the systemic level of inflammation and can be of clinical use before chemotherapy treatment, during treatment, and for determining life expectancy [12,13].

This study aimed to investigate the relationship between the above-mentioned biomarkers and the survival outcomes in colorectal cancer patients undergoing FOLFOX/CAPOX or FOLFIRI/CAPIRI chemotherapy regimes, including targeted therapies, specifically anti-EGFR or anti-VEGF agents. We divided a cohort of 188 patients into groups based on whether they were alive or deceased as of a specific cutoff date, namely 1 March 2023. The aim was to explore how these biomarkers, reflecting the body’s immune and inflammatory responses, correlate with patient survival. By analyzing these markers, we sought insights that could potentially guide more effective management and treatment approaches in colorectal cancer care.

## 2. Materials and Methods

### 2.1. Criteria

The study’s retrospective cohort was carefully selected from the Electronic Medical Records at the OncoHelp Association in Timisoara after obtaining approval from the Institutional Review Board. The period examined stretched from 1 January 2016 to 1 March 2023. The selection process was anchored on key criteria such as age, gender, and specific details related to CRC like its stage, grade, the treatments patients received, and important biomarkers including RAS mutation status, and levels of hemoglobin (HGB), white blood cells (WBC), lymphocytes, neutrophils, and platelets.

Out of an initial pool, a total of 1547 patients with advanced stages of CRC were found. The selection process led to the exclusion of 992 patients due to their CRC being at stages I-III and lacking hepatic metastasis progression. Furthermore, 266 patients were excluded from undergoing treatments outside the study’s specified regimens (FOLFOX/CAPEOX or FOLFIRI/CAPIRI in combination with targeted therapies such as anti-VEGF (Bevacizumab) and anti-EGFR treatments (Cetuximab or Panitumumab)), ensuring consistency in the treatment approaches analyzed. The cohort was further refined by excluding 75 individuals with dual neoplasias, 21 due to incomplete medical records, and 5 for ambiguous hepatic metastasis status (Figure 1).

The final cohort consisted of 188 patients and was divided into two groups based on their survival status as of 1 March 2023: those who were still living (Arm A) and those who had deceased (Arm B). Within these groups, the study calculated various indices and ratios to explore the relationship between physiological markers and patient outcomes. This included the Body Mass Index (BMI), calculated using the standard formula (kg/m^2^). The NHL score was computed from neutrophil counts divided by the product of hemoglobin levels and lymphocyte count (N/(H × L)). Additionally, the study looked at the Neutrophils to Lymphocytes Ratio and the Platelets to Lymphocytes Ratio, calculated as the neutrophil count to lymphocyte count (N/L) and platelet count to lymphocyte count (P/L), respectively. The Systemic Immune-Inflammation Index was determined using the formula: platelet count times the ratio of neutrophil count to lymphocyte count (P × (N/L)). It is important to note that these biological parameters are based on mean values from the last 4–6 hospital visits, providing a solid overview of each patient’s physiological condition.

### 2.2. Statistical Analysis

In the statistical analysis subsection of the article focused on 188 colorectal cancer patients with hepatic metastasis, the methodologies employed to analyze patient demographics, clinical characteristics, and prognostic factors are meticulously outlined, ensuring clarity and rigor in the presentation of the research findings.

The descriptive analysis of the study population was conducted using mean, standard deviation (SD), and 95% confidence interval (CI) used for normally distributed continuous variables. Categorical variables were summarized using frequencies and percentages. This approach provided a comprehensive overview of the patient demographics and baseline clinical characteristics, facilitating a nuanced understanding of the study cohort.

For univariate analysis, the study employed various statistical tests to examine the association between clinical characteristics and patient survival. The t-test was used for continuous variables to compare means between two independent groups, while the Chi-square test was utilized for categorical variables to assess the distribution of frequencies. This enabled the identification of potential prognostic factors and their impact on patient outcomes.

Logistic regression analysis was conducted to further explore the relationship between BMI, Lymphocytes, NHL score, NLR, PLR, and SII, and the odds of mortality. This analysis provided odds ratios (ORs) along with 95% confidence intervals, offering insights into the strength and direction of these associations. Variables that demonstrated significance in univariate analysis were included in the logistic regression model, facilitating a more precise understanding of factors influencing mortality.

The Kaplan-Meier method was used for survival analysis, with the log-rank test employed to compare survival curves between sex, RAS, FOLFOX/CAPEOX, FOLFIRI/CAPIRI, and metastases at diagnosis. This method allowed for the estimation of median survival times and the examination of survival distributions across various patient subgroups, highlighting differences in prognosis.

ROC curve analysis was conducted to evaluate the diagnostic accuracy of NHL score, NLR, PLR, and SII in predicting patient outcomes. The Area Under the Curve (AUC), along with sensitivity and specificity values, was calculated to assess the performance of these markers in distinguishing between different patient outcomes.

Statistical significance was set at a *p*-value of <0.05 for all tests, and all analyses were performed using MedCalc Statistical Software version 20.218 (MedCalc Software Ltd., Ostend, Belgium; https://www.medcalc.org, accessed on 26 May 2024; 2023). This rigorous statistical framework ensured the reliability and validity of the study findings, contributing to the body of knowledge on CRC with hepatic metastasis and informing clinical practice and future research directions.

## 3. Results

### 3.1. Participant Clinical Characteristics

In the investigation of CRC patients with hepatic metastasis, a comprehensive descriptive analysis of patient demographics and clinical characteristics was meticulously conducted. This encompassed a total of 188 participants. Age distribution within this cohort was characterized by a mean of 61.3 years (SD ± 10.5), with 95% CI from 59.7 to 62.8 years, reflecting a range between 29 and 83 years. BMI revealed an average of 25.5 (SD ± 5.30), with 95% CI between 24.7 and 26.2, and observed values ranging from 14.0 to 46.6 (Table 1).

Regarding hematological parameters, the mean hemoglobin level was recorded at 12.0 g/dL (SD ± 1.76), with a 95% CI from 11.7 to 12.2 g/dL, showcasing a spectrum from 7.30 to 18.0 g/dL. The WBC count stood at an average of 7.78 × 10^9^/L (SD ± 3.48), with 95% CI between 7.28 and 8.28 × 10^9^/L, and ranged from 3.33 to 28.1 × 10^9^/L. Lymphocyte counts averaged at 1.75 × 10^9^/L (SD ± 0.641), with 95% CI from 1.65 to 1.84 × 10^9^/L, extending from 0.653 to 4.76 × 10^9^/L. Neutrophil counts were observed at a mean of 5.16 × 10^9^/L (SD ± 3.14), with 95% CI ranging from 4.71 to 5.61 × 10^9^/L and a range of 1.74 to 24.1 × 10^9^/L. The platelet count exhibited an average of 255 × 10^9^/L (SD ± 100.0), with 95% CI between 240.9 and 269.7 × 10^9^/L, spanning from 49.0 to 583 × 10^9^/L.

The NHL score, a novel marker of interest within this study, averaged at 0.293 (SD ± 0.234), with a 95% CI of 0.26 to 0.33 and values ranging from 0.0654 to 1.49. NLR and PLR were also thoroughly analyzed, presenting means of 3.29 (SD ± 2.25) with 95% CI between 2.97 and 3.61 and 163 (SD ± 84.3) with 95% CI from 151.1 to 175.3, respectively. SII was detailed with a mean of 955 (SD ± 1045) and a 95% CI from 804.7 to 1105.6, delineating the breadth of immune-inflammatory response within this cohort.

### 3.2. Univariate Analysis

In the univariate analysis segment of the study focusing on colorectal cancer (CRC) patients with hepatic metastasis, the comparison between arm A/surviving patients (n = 96) and arm B/deceased patients (*n* = 92) unveiled notable findings across various clinical parameters.

Age did not present a significant difference between the two groups, with surviving patients having a mean age of 62.1 years (SD ± 10.5) compared to 60.4 years (SD ± 10.5) for the deceased, *p*-value = 0.25. Gender distribution showed a slight, non-significant skew towards a higher percentage of females in the deceased group (46%) compared to the living (36%), with a *p*-value of 0.2.

Interestingly, BMI appeared to play a role, with survivors having a higher mean BMI of 26.3 (SD ± 5.53) versus 24.6 (SD ± 4.93) for the deceased, reaching statistical significance with a *p*-value of 0.036.

The lymphocyte count was significantly higher in surviving patients, averaging 1.85 (SD ± 0.661) compared to 1.64 (SD ± 0.605) in the deceased group, with a *p*-value of 0.028. However, neutrophil counts, hemoglobin levels, WBC, and platelet counts did not significantly differ between the two groups.

The NHL score was significantly higher in the deceased group (mean 0.339, SD ± 0.267) compared to survivors (mean 0.249, SD ± 0.187), with a *p*-value < 0.01, highlighting its potential as a prognostic marker. The mean NLR was also found to be significantly different between the two groups, with values of 2.83 (±1.76) in the alive group and 3.78 (±2.58) in the deceased group, indicating a higher degree of systemic inflammation in the latter (*p* < 0.01).

PLR and SII also demonstrated significant differences, with higher values in deceased patients, with a *p* = 0.048 for PLR and *p* = 0.027 for SII, further emphasizing the importance of inflammatory markers in CRC prognosis. The differences in these parameters underscore the potential role of systemic inflammation and immune response in the progression and outcome of CRC with hepatic metastasis (Table 2).

### 3.3. Logistic Regression Analysis

The logistic regression analysis in this study offers insightful revelations into the relationship between clinical characteristics and the likelihood of mortality. Each variable’s odds ratio, confidence interval, and statistical significance elucidate their impact on patient outcomes.

BMI emerged as a significant factor, with an OR of 0.9416 (95% CI: 0.8893 to 0.9971), suggesting a slight protective effect against mortality for each unit increase in BMI, achieving statistical significance (*p* = 0.0394). This inverse relationship underscores the complex interplay between body composition and CRC prognosis (Table 3).

Lymphocyte counts also held prognostic value, with an OR of 0.5891 (95% CI: 0.3633 to 0.9553), indicating that higher lymphocyte levels are associated with a reduced risk of death (*p* = 0.0319). This finding highlights the crucial role of the immune system in combating CRC progression (Figure 2).

The NHL score, with an OR of 6.1071 (95% CI: 1.5097 to 24.7048), pointed to a significant increase in the risk of mortality with higher scores (*p* = 0.0112), emphasizing its potential as a prognostic marker for CRC patients.

NLR also proved to be a significant predictor of mortality, with an OR of 1.2269 (95% CI: 1.0613 to 1.4183), reinforcing the concept that elevated inflammatory markers are indicative of poorer outcomes (*p* = 0.0057).

PLR and SII were both examined, yet only the PLR approached statistical significance (*p* = 0.0511), suggesting a potential, albeit less robust, association with CRC mortality.

### 3.4. Kaplan-Meier Survival Analysis

The Kaplan-Meier survival analysis clarifies the impact of various factors on overall survival (OS), providing a nuanced understanding of their prognostic significance (Table 4).

Gender appeared to influence OS, with females having a mean OS of 27.1 months (95% CI: 22.1 to 32) and males exhibiting a longer mean OS of 29.6 months (95% CI: 26.3 to 35.8), although the difference did not reach statistical significance (*p* = 0.07). This suggests potential underlying biological or treatment-related differences between genders that warrant further exploration.

The presence of RAS mutations was investigated, revealing that patients with mutant RAS had a mean OS of 31 months (95% CI: 22.5 to 34.1), compared to 28.3 months (95% CI: 19.2 to 28.4) for those with wild-type RAS (*p* = 0.5853).

Treatment regimens augmented with targeted therapies, such as anti-VEGF (Bevacizumab) and anti-EGFR treatments (Cetuximab or Panitumumab), were also analyzed. Patients receiving FOLFOX/CAPEOX in combinations with these targeted agents had a mean OS of 31.8 months (95% CI: 26.3 to 37.3), while those on FOLFIRI/CAPIRI (alongside the same targeted therapies) had a slightly lower mean OS of 29.5 months (95% CI: 24.3 to 34.7), although these differences were not statistically significant. This indicates that, within this cohort, treatment type did not drastically alter survival outcomes.

The presence of metastases at diagnosis was a critical factor, with patients having no metastases at diagnosis showing a significantly higher mean OS of 48.9 months (95% CI: 38.8 to 59.1), compared to 26.4 months for those with metastases (95% CI: 22.9 to 34.5), with a significant *p*-value of 0.0001. This stark contrast underscores the profound impact of metastatic disease at diagnosis on CRC prognosis (Figure 3).

### 3.5. Receiver Operating Characteristic Curve

The Neutrophil-Hemoglobin-Lymphocyte (NHL) score demonstrated an AUC of 0.625 (95% CI: 0.545 to 0.705), with a statistically significant *p*-value of 0.0022. The optimal threshold for NHL score was determined to be >0.21, yielding a sensitivity of 60.87% and a specificity of 61.46%. This indicates a moderate predictive value of the NHL score in distinguishing between patient outcomes, suggesting its potential utility in clinical decision-making (Figure 4).

The Neutrophil–Lymphocyte Ratio (NLR) also showed a promising AUC of 0.622 (95% CI: 0.542 to 0.702), with a significant *p*-value of 0.0027. The threshold for NLR was established at >2.86, providing a sensitivity of 57.61% and a specificity of 66.67%. This underscores the NLR’s potential as a prognostic marker, albeit with limitations in sensitivity and specificity that warrant cautious interpretation.

The Platelet-Lymphocyte Ratio (PLR) yielded an AUC of 0.594 (95% CI: 0.512 to 0.675), with a *p*-value of 0.0237. A PLR threshold of >91.8 achieved high sensitivity (90.22%) but relatively low specificity (29.17%), indicating its potential for identifying patients at risk, though it may generate a higher rate of false positives.

The Systemic Immune-Inflammation Index (SII) presented an AUC of 0.607 (95% CI: 0.526 to 0.688), with a *p*-value of 0.0094. The optimal SII threshold was identified as >340.4, resulting in a sensitivity of 89.13% and a specificity of 32.29%. The high sensitivity suggests SII’s utility in screening, but its lower specificity highlights the need for confirmatory tests (Table 5).

## 4. Discussions

Our retrospective cohort study revealed notable differences between the survival outcomes of colorectal cancer patients based on several clinical and laboratory parameters. The significant factors identified include BMI, lymphocyte counts, NLR, PLR, SII, and the NHL score. Each of these factors underwent rigorous statistical evaluation to ascertain their prognostic significance.

The analysis revealed a statistically significant difference in BMI between the deceased (mean ± SD: 24.6 ± 4.93) and alive (mean ± SD: 26.3 ± 5.53) cohorts, with a *p*-value of 0.036, indicating a potential link between lower BMI and increased mortality in CRC patients. This observation was further substantiated through logistic regression analysis, where BMI presented an odds ratio of 0.9416, with a 95% confidence interval ranging from 0.8893 to 0.9971 and a *p*-value of 0.0394, suggesting that each unit decrease in BMI is associated with a higher risk of death. This aligns with the notion of the “obesity paradox” in cancer, where higher BMI has been associated with improved survival in certain cancers, suggesting a protective role of nutritional reserves during the course of the disease and treatment [14,15,16].

A recent meta-analysis by Li et al. not only corroborates our findings but also enriches our discussion by providing a broader context and suggesting potential biological mechanisms that could underlie the observed association between BMI and CRC mortality. This underscores the multifaceted nature of the obesity paradox in CRC and highlights the importance of leptin upregulation, with its influence on VEGF and VEGF receptor-2 (suggesting that they make tumors more susceptible to tyrosine kinase inhibitors), and adipokine downregulation, with its anti-inflammatory-and-anti-proliferative-effects [17]. However, the body composition data presented in the study by Caan et al., particularly the protective role of adequate muscle reserves in patients with a BMI in the overweight range, challenge the adequacy of BMI as a standalone metric for assessing survival risk in CRC patients. It suggests that the lower mortality associated with higher BMI may be attributable to better muscle mass rather than adiposity per se. One of the key mechanisms highlighted is the nutrient mobilization from skeletal muscle to the tumor. This process involves the direct use of amino acids from skeletal muscle and the indirect utilization of glucose derived from liver gluconeogenesis, facilitated by the tumor’s influence on the body’s metabolism [18]. At the same time, our study contributes to the ongoing discussion on the obesity paradox in CRC. The study mentioned above highlights the need for a nuanced understanding of the factors influencing patient outcomes, particularly the critical role of sarcopenia.

The prognostic value of lymphocyte counts in different cancer types has garnered increasing attention in recent oncological research. In our study, we observed that lymphocyte counts were significantly lower in the deceased group (mean ± SD: 1.64 ± 0.605) compared to those who were alive (mean ± SD: 1.85 ± 0.661), with a *p*-value of 0.028. This finding is in concordance with the logistic regression analysis, which illustrated an OR of 0.5891 (95% CI: 0.3633 to 0.9553) with a *p*-value of 0.0319, thereby underscoring the inverse relationship between lymphocyte counts and mortality risk in CRC patients. Parallel observations have been noted in breast and lung cancer research, emphasizing the universality of lymphocytes as a prognostic marker. The findings indicated a strong association between low lymphocyte counts and poorer patient outcomes, suggesting that a reduced percentage of lymphocytes in the peripheral blood could serve as a valuable predictor of adverse prognosis in breast and lung cancer [19,20], but also in colorectal cancer.

The analysis of our retrospective cohort study highlights the potential prognostic value of the NHL score in CRC, revealing its significant association with mortality. The NHL score, which was significantly higher in the deceased cohort (mean ± SD: 0.339 ± 0.267) compared to those alive (mean ± SD: 0.249 ± 0.187) with a *p*-value < 0.01, showed a pronounced association with mortality through logistic regression analysis (OR: 6.1071, 95% CI: 1.5097 to 24.7048, *p*-value: 0.0112). To date, we have not found any studies in the literature that specify the use of NHL scores to determine prognosis in CRC. However, the NHL score was correlated with inflammatory states and immune status in rheumatoid arthritis and cardiovascular disease [21,22]. Moreover, we found a study that delved into the predictive value of the NHL score, among other hematological indices, in relation to postoperative recurrence of non-muscle invasive bladder cancer (NMIBC). The analysis identified the NHL score as an independent risk factor for predicting cancer recurrence [23]. The evidence suggesting the NHL score’s predictive value in NMIBC, coupled with its association with mortality in our CRC cohort, advocates for its potential utility as a prognostic marker in cancer. Further exploration is warranted to validate the NHL score’s utility in CRC and to potentially integrate it into clinical practice, enhancing our ability to predict disease outcomes and tailor treatments more effectively.

In our study, the NLR stood out for its associations with patient outcomes. NLR, with a HR of 1.2269 and a significant *p*-value of 0.0057, suggests a higher NLR could be indicative of a poorer prognosis in CRC, pointing to the potential of NLR as a valuable prognostic marker. These findings are consistent with previous studies that have identified high NLR values as indicators of systemic inflammation and poor prognosis in CRC patients [24,25]. A recent meta-analysis, comprising 32.788 patients, investigated the prognostic utility of pre-treatment NLR for overall survival and progression-, recurrence-, or disease-free survival in CRC patients, claiming that NLR can be fully assessed as an independent predictor of CRC progression and outcome further supporting our finding [26].

On the other hand, SII and PLR, despite their slight association with CRC outcomes, present a minimal effect size, hinting at its limited clinical relevance in our study compared to NLR. A possible explanation comes from a study by Hernandez-Ains et al., revealing that right-sided colon cancer patients exhibited significantly higher values of PLR and SII compared to left-sided colon cancer and rectal cancer, suggesting a possible variance in the inflammatory response based on tumor location [27]. Our study cohort was formed predominantly from patients diagnosed with distal CRC (N = 149) and a small group of proximal CRC (N = 39). Further study is essential for a thorough understanding of PLR and SII regarding CRC cancer localization and prognosis.

The Kaplan-Meier survival analysis conducted in our analysis indicated a difference in mean OS between females (27.1 months, 95% CI: 22.1 to 32) and males (29.6 months, 95% CI: 26.3 to 35.8), while the HR for females, when compared to males, was 1.5030 (95% CI: 0.9601 to 2.3527), indicating a trend towards higher risk in females, but without reaching statistical significance (*p* = 0.07). Our observation of a trend toward mortality risk in females, though not reaching statistical significance, mirrors broader discussions in the literature. A relevant example is a study featured in BMC Cancer. This study notes the higher participation of women in screening programs (in the United Kingdom), yet paradoxically, men have a higher likelihood of early-stage CRC detection via these screenings, correlating to more optimistic survival prospects. The review further underscores the unfortunate pattern of later-stage CRC identification in women, which invariably leads to poorer survival outcomes [28].

A significant finding from our analysis was the impact of metastases on OS upon diagnosis. Patients without metastases had a notably longer mean OS of 48.9 months (95% CI: 38.8 to 59.1) compared to those with metastases, who had a mean OS of 26.4 months (95% CI: 22.9 to 34.5), with a *p*-value of 0.0001. The HR for patients with metastases was 2.6313 (95% CI: 1.6437 to 4.2120), underscoring the substantial impact of metastatic disease on prognosis. This highlights the critical importance of early detection and intervention in CRC to improve survival outcomes, aligning with evidence that metastatic disease is a pivotal determinant of prognosis in CRC [29].

Regarding the AUC values, sensitivity, specificity, and the optimal cut-off points of the predictive performance of various biomarkers in CRC prognosis, the NHL score demonstrated a significant prognostic utility with an AUC of 0.625 (95% CI: 0.545 to 0.705, *p*-value: 0.0022), indicating a moderate predictive performance. The optimal criterion identified for the NHL score was >0.21, yielding a sensitivity of 60.87% and a specificity of 61.46%. To date, a thorough review of the literature reveals a notable absence of studies investigating the use of NHL scores in the context of metastatic CRC. This gap contrasts with the insights collected from the research conducted by Kim et al. and Zhao et al., which elucidate the prognostic value of the NHL score in cardiovascular diseases and non-muscle-invasive bladder cancer, with optimal cutoff values of >0.35 for major adverse cerebrocardiovascular events, respectively >252.645 for non-muscle-invasive bladder cancer (to mention that they used another formula to calculate the NHL score) [22,23]. Our findings align with these studies, highlighting the NHL score’s potential as a further prognostic tool, even for CRC. The sensitivity and specificity, while not highly precise, offer a reasonable initial screening tool for identifying CRC patients at higher risk of mortality. However, the variability in optimal thresholds across different studies underscores the context-specific nature of the NHL score’s utility, necessitating tailored formulas and cutoff values that accommodate the diverse pathophysiological and clinical nuances of various malignancies/diseases. This collective body of work underscores the unexplored potential of the NHL score in mCRC, inviting further exploration to bridge this significant gap in the literature.

The NLR achieved a comparable AUC of 0.622 (95% CI: 0.542 to 0.702, *p*-value: 0.0027), with an optimal threshold of >2.86. This threshold resulted in a sensitivity of 57.61% and a specificity of 66.67%, indicating a slightly better performance in correctly identifying patients with worse prognoses than in avoiding false positives. The NLR’s utility in CRC prognosis, especially given its simple calculation and widespread availability, aligns with the existing literature emphasizing the role of systemic inflammation in cancer progression. Vano et al. provided detailed analyses across a varied solid tumor patient cohort to pinpoint an “optimal” NLR cut-off range of 3.5 to 4.5 before treatment, associating higher NLR values with a marked rise in mortality risk. Also, the study highlights the critical nature of incorporating a time-dependent component in NLR evaluations, which unveiled a more pronounced effect over time [30]. Another retrospective study that analyzed the NLR as a prognostic biomarker in mCRC sets a lower NLR cut-off value at >2.35. [31] This study suggests the dynamic character of NLR as a prognostic tool, reinforcing the findings presented in our study.

Our findings demonstrated a high sensitivity of PLR in our study, at 90.22% with a specificity of 29.17%, suggesting its potential in identifying at-risk patients. Similarly, the SII’s high sensitivity of 89.13%, despite low specificity (32.29%), underscores its possible role in risk stratification. These variations in sensitivity and specificity in PLR and SII may be attributed to differences in biomarkers associated with tumor location, as noted in the existing literature [27,32]. This discrepancy underscores the necessity for a nuanced understanding and application of these biomarkers in CRC, taking into consideration factors such as tumor stage, location (proximal/distal), patient demographics, presence of comorbid conditions, and treatments, which can all influence biomarker levels.

Building on the current findings, future investigations could significantly advance our understanding of biomarkers in CRC. Prospective cohort studies are particularly recommended to validate the prognostic significance of biomarkers like BMI, lymphocyte counts, and inflammatory indices such as NLR, PLR, SII, and the NHL score. Moreover, in-depth genetic and molecular studies underlying E-Cadherin dysregulation in these cancers, particularly considering the loss of E-Cadherin in various highly malignant and dedifferentiated cancers such as Chondrosarcoma and small cell carcinomas of the urinary bladder, as well as for reduced E-Cadherin expression like invasive lobular breast cancer and colorectal cancer [33,34], could elucidate the pathways through which these biomarkers interact with CRC progression. Another promising area involves exploring the impact of tumor stage and location on biomarker efficacy, potentially tailoring prognostic tools to enhance precision in clinical settings. Lastly, integrating advanced statistical models and machine learning could uncover complex interactions between biomarkers, offering a more robust predictive capability for CRC outcomes. Each of these research avenues holds the potential to refine prognostic assessments and guide more personalized treatment strategies for CRC patients.

## 5. Conclusions

Our retrospective cohort study has made significant contributions to the understanding of prognostic biomarkers in colorectal cancer, affirming the complexity of the disease and the multifaceted nature of its prognosis. The analysis has emphasized the nuanced implications of various clinical and laboratory parameters such as BMI, lymphocyte counts, and different hematological indices, including NLR, PLR, SII, and the NHL score.

Firstly, our study corroborates the intriguing concept of the “obesity paradox”, where a higher BMI appears to confer a protective effect, potentially due to better nutritional reserves. However, the concurrent observation of the importance of muscle mass over adiposity suggests that BMI alone may be an inadequate metric for assessing survival risk in CRC patients. This calls for a more holistic approach to patient assessment, possibly integrating muscle mass measurements to better predict outcomes.

Furthermore, our findings concerning the prognostic value of lymphocyte counts and the NHL score have added to the growing body of evidence that suggests systemic inflammation and immune competence play critical roles in cancer progression and patient survival. These biomarkers have demonstrated significant prognostic utility, indicating poorer outcomes associated with lower lymphocyte counts and higher NHL scores.

The prognostic implications of NLR, PLR, and SII also merit attention, although their varied performance based on tumor location and other patient-specific factors like disease stage and comorbid conditions suggests the need for a contextual application of these markers. This variability in biomarker effectiveness highlights the necessity for ongoing research and tailored clinical strategies to refine their prognostic capabilities.

In conclusion, our study has expanded the current understanding of prognostic factors in CRC, advocating for a multi-dimensional approach to prognostic evaluations. This approach should consider not only the traditional metrics such as tumor stage and histological grading but also incorporate a broader spectrum of biomarkers. Future studies should aim to validate these findings and explore the integration of these biomarkers into routine clinical practice, enhancing the precision of prognostic assessments and ultimately guiding more personalized treatment strategies for CRC patients. This exploration will undoubtedly contribute to improving overall survival outcomes and quality of life for CRC patients.

### Limitations

This study, while providing valuable insights into the prognosis of colorectal cancer patients with hepatic metastasis, has several limitations that merit consideration. First, the retrospective cohort design is limited by the specific inclusion criteria, focusing on advanced CRC stages and certain treatment regimens.

Another significant limitation is the potential for confounding factors that were not controlled for or identified during the study. Variables such as socioeconomic status, lifestyle choices, and other genetic factors beyond RAS mutation status could influence both the prognosis and the effectiveness of the treatments, yet were not accounted for in the analyses. The sample size, though adequate for basic statistical analyses, may lack the statistical power necessary to detect small but clinically significant effects, especially in subgroup analyses or interaction effects between treatments and patient demographics or biomarkers.

Additionally, while the use of survival analysis techniques like the Kaplan-Meier method provides a robust tool for understanding survival patterns, these methods assume proportional hazards—an assumption that may not always hold true across different patient subgroups or over extended periods.

The diagnostic accuracy of biomarkers such as the NHL score, NLR, and others used in this study also poses limitations. The moderate sensitivity and specificity values indicate that these biomarkers, while helpful, are not definitive for clinical decision-making without additional confirmatory tests.

Moreover, the reliance on specific chemotherapy and targeted therapy regimens limits the applicability of the findings to these treatments alone. As treatment modalities for CRC evolve, the relevance of these results may diminish unless continuously updated.

These limitations underscore the need for a cautious interpretation of the findings and suggest that further studies, preferably prospective and involving a broader spectrum of patients and treatments, are required to confirm these results and expand our understanding of CRC management.

## Figures and Tables

**Figure 1 medicina-60-01003-f001:**
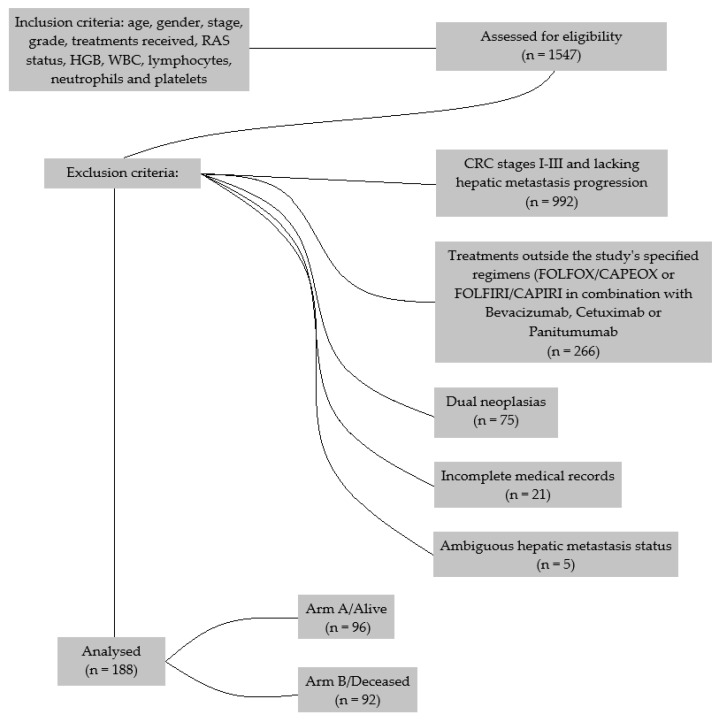
Consort Flow Diagram for the inclusion and the reasons for exclusion of all the samples.

**Figure 2 medicina-60-01003-f002:**
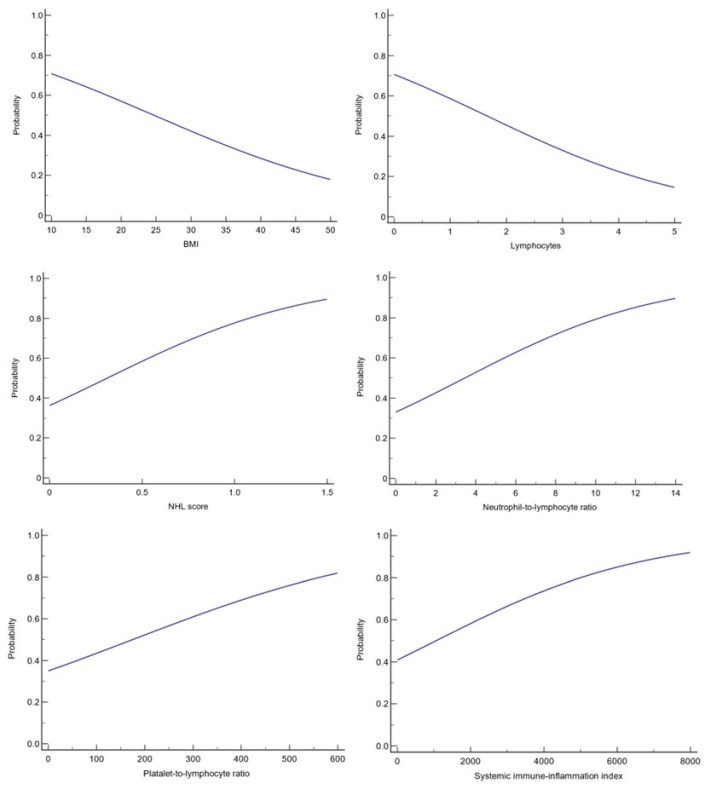
Logistic regression for significance.

**Figure 3 medicina-60-01003-f003:**
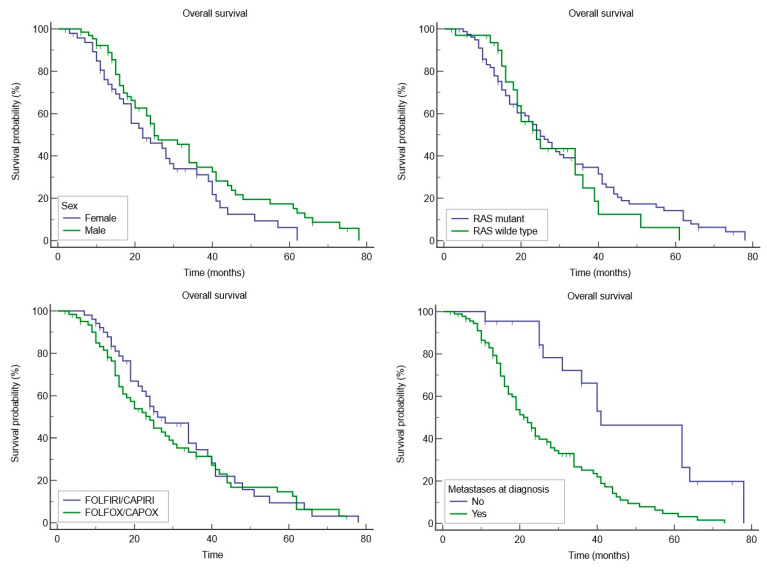
Kaplan-Meier survival analysis.

**Figure 4 medicina-60-01003-f004:**
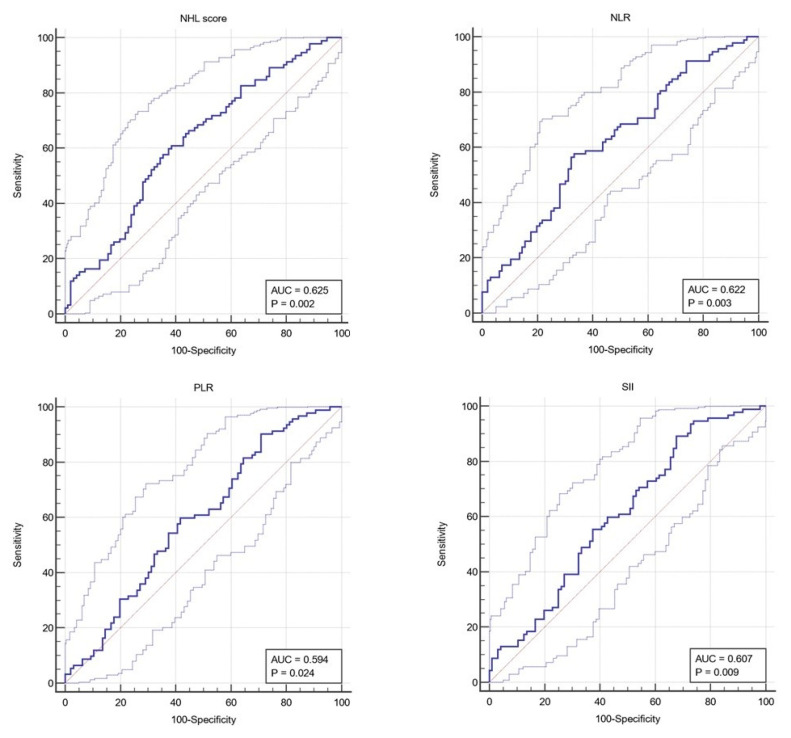
ROC curve. Dark blue curve represents the true positive rate (sensitivity) versus the false positive rate (100-specificity) at various threshold settings; Diagonal line/reference line serves as a baseline (AUC = 0.5); Upper and lower confidence interval indicates the best and the worst expected of the classifier within the 95% confidence interval.

**Table 1 medicina-60-01003-t001:** Participant Clinical Characteristics.

	Mean (±SD)	95% CI	Min	Max	*n*
Age	61.3 (±10.5)	59.7 to 62.8	29.0	83.0	188
BMI	25.5 (±5.30)	24.7 to 26.2	14.0	46.6	188
Haemoglobin	12.0 (±1.76)	11.7 to 12.2	7.30	18.0	188
WBC	7.78 (±3.48)	7.28 to 8.28	3.33	28.1	188
Lymphocytes	1.75 (±0.641)	1.65 to 1.84	0.653	4.76	188
Neutrophils	5.16 (±3.14)	4.71 to 5.61	1.74	24.1	188
Platelets	255 (±100.0)	240.9 to 269.7	49.0	583	188
NHL score	0.293 (±0.234)	0.26 to 0.33	0.0654	1.49	188
NLR	3.29 (±2.25)	2.97 to 3.61	0.924	12.3	188
PLR	163 (±84.3)	151.1 to 175.3	17.5	519	188
SII	955 (±1045)	804.7 to 1105.6	52.5	7150	188

**Table 2 medicina-60-01003-t002:** Logistic regression between arm A (alive) and arm B (deceased).

		Alive (N = 96) Mean (±SD)	Deceased (N = 92) Mean (±SD)	N (%)	*p*-Value
Age		62.1 (±10.5)	60.4 (±10.5)	188	0.25
Sex	Female	35 (36%)	42 (46%)	77 (41%)	0.2
	Male	61 (64%)	50 (54%)	111 (59%)	-
BMI		26.3 (±5.53)	24.6 (±4.93)	188	0.036
HGB		12.1 (±1.80)	11.8 (±1.71)	188	0.19
WBC		7.48 (3.28)	8.10 (3.67)	188	0.22
Lymphocytes		1.85 (±0.661)	1.64 (±0.605)	188	0.028
Neutrophiles		4.78 (±2.87)	5.56 (±3.36)	188	0.091
Platelets		248 (97.2)	263 (103)	188	0.33
NHL score		0.249 (±0.187)	0.339 (±0.267)	188	<0.01
NLR		2.83 (±1.76)	3.78 (±2.58)	188	<0.01
PLR		151 (79.7)	176 (87.5)	188	0.048
SII		789 (746)	1129 (1267)	188	0.027
CRC	proximal	25 (26%)	14 (15%)	39 (21%)	0.067
	distal	71 (74%)	78 (85%)	149 (79%)	-
FOLFIRI/CAPIRI		32 (33%)	38 (41%)	70 (37%)	0.26
FOLFOX/CAPOX		64 (67%)	54 (59%)	118 (63%)	-
Grade	G1	7 (7.3%)	7 (7.6%)	14 (7.4%)	0.52
	G2	83 (86%)	75 (82%)	158 (84%)	-
	G3	6 (6.2%)	10 (11%)	16 (8.5%)	-
Metastases at diagnosis	No	13 (14%)	14 (15%)	27 (14%)	0.74
	Yes	83 (86%)	78 (85%)	161 (86%)	-
RAS	mutant	68 (71%)	69 (75%)	137 (73%)	0.52
	WILDE TYPE	28 (29%)	23 (25%)	51 (27%)	-

**Table 3 medicina-60-01003-t003:** Logistic regression for significant factors in univariable analysis.

	OR	95% CI	Coefficient	Std. Error	Wald	*p*-Value	AUC
BMI	0.9416	0.8893 to 0.9971	−0.060130	0.029183	4.2456	0.0394	0.585
Lymphocytes	0.5891	0.3633 to 0.9553	−0.52908	0.24662	4.6025	0.0319	0.598
NHL score	6.1071	1.5097 to 24.7048	1.80946	0.71303	6.4399	0.0112	0.625
NLR	1.2269	1.0613 to 1.4183	0.20447	0.073959	7.6431	0.0057	0.622
PLR	1.0036	1.0000 to 1.0071	0.0035474	0.0018189	3.8036	0.0511	0.594
SII	1.0003	1.0000 to 1.0007	0.00034991	0.00016521	4.4856	0.0342	0.607

**Table 4 medicina-60-01003-t004:** Kaplan-Meier survival analysis.

		Mean OS (Months)	95% CI	*p*-Value	HR (95% CI)
Sex	Female	27.1	22.1 to 32	0.07	1.5030 [0.9601 to 2.3527]
	Male	29.6	26.3 to 35.8		0.6653 [0.4250 to 1.0415]
RAS	Mutant	31	22.5 to 34.1	0.5853	0.8654 [0.5150 to 1.0454]
	Wilde-type	28.3	19.2 to 28.4		1.1555 [0.6876 to 1.9419]
FOLFOX/CAPOX		31.8	26.3 to 37.3	0.5777	1.1293 [0.7360 to 1.7327]
FOLFIRI/CAPIRI		29.5	24.3 to 34.7		0.8855 [0.5771 to 1.3587]
Metastases at diagnosis	No	48.9	38.8 to 59.1	0.0001	0.3800 [0.2374 to 0.6084]
	Yes	26.4	22.9 to 34.5		2.6313 [1.6437 to 4.2120]

**Table 5 medicina-60-01003-t005:** ROC curve.

	AUC	95% CI	*p*-Value	Associated Criterion	Sensitivity	Specificity
NHL SCORE	0.625	0.545 to 0.705	0.0022	>0.21	60.87	61.46
NLR	0.622	0.542 to 0.702	0.0027	>2.86	57.61	66.67
PLR	0.594	0.512 to 0.675	0.0237	>91.8	90.22	29.17
SII	0.607	0.526 to 0.688	0.0094	>340.4	89.13	32.29

## Data Availability

The data generated or analyzed during this study are included in this published article or are available from the corresponding author upon reasonable request.

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
