# Peer review of "Influence of Biomarkers on Mortality among Patients with Hepatic Metastasis of Colorectal Cancer Treated with FOLFOX/CAPOX and FOLFIRI/CAPIRI, Including Anti-EGFR and Anti-VEGF Therapies"

_medicina, 2024, doi:10.3390/medicina60061003_

Round 1

Reviewer 1 Report

Comments and Suggestions for Authors

This is a great paper that provides valuable insights into the prognosis of retrospective cohorts of colorectal cancer patients with hepatic metastasis.  The study has elaborated on the current insight of prognostic factors in CRC, including incorporation of  a broader spectrum of biomarkers promoting for a multi-dimensional approach to prognostic evaluations.

Strengths:

- The references are most recent and comprehensive.   
- Interpretation and presentation of previous studies is accurate.

- Tables and figures are  thorough.
- No major suggestions for improvements.          
- Clarity and context in this paper are good.

Author Response

Dear Reviewer,
Thank you very much for your thoughtful and positive feedback on our paper. We are delighted to hear that you found our study valuable and insightful regarding the prognosis of retrospective cohorts of colorectal cancer patients with hepatic metastasis. Your recognition of the strengths in our research, including the comprehensive references, accurate presentation of previous studies, and thoroughness of our tables and figures, is greatly appreciated.
We are particularly pleased that you found the clarity and context of the paper to be good, and that you see no major suggestions for improvements. Your encouraging comments serve as a testament to our efforts and dedication in contributing to the understanding of prognostic factors in colorectal cancer.
Once again, thank you for your time and constructive feedback.

Reviewer 2 Report

Comments and Suggestions for Authors

In this study, a retrospective cohort of 188 patients with colorectal cancer (CRC) and liver metastases was analysed for demographic data, clinical characteristics and biomarkers such as lymphocyte counts and various inflammatory indices. CRC patients were selected from those undergoing FOLFOX/CAPOX or FOLFIRI/CAPIRI chemotherapy, including targeted therapies, especially anti-EGFR or anti-VEGF agents. The work is worth adding to the literature and still needs minor corrections.

In the criteria section, the cut-off date is stated as 3 March 2023, but in the previous paragraph it is 01/03/2023. This should be corrected.

CAPEOX is misspelled in Table 2.

A higher mean BMI associated with survivors is an open topic for discussion. Although the p-value is 0.036, the data are not very different from each other to make this speculative comment based on obesity open to discussion. The authors would do better to explain this with references to other similar studies. The references given 9-11 in the discussion do not explain it well enough.

The authors speculate that the NHL score may be a prognostic marker. They should better explain the scientific basis of this calculation to increase the impact of using it as a prognostic marker.

As this point has only recently been discussed and is also included in the study, the dihydropirimidine dehydrogenase test data of any patients should be presented in a table if there is data for the included cohort. This adds a new dimension to the article.

The comment on the association between chemotherapy and observed biomarkers is not sufficient and needs to be detailed.

Author Response

Dear reviwer, here are the corrections and clarifications:

  • In the criteria section, the cut-off date is stated as 3 March 2023, but in the previous paragraph, it is listed as 01/03/2023. This has been corrected to consistently state 01 March 2023 throughout the document.
  • The term "CAPEOX" in Table 2 was misspelled. This has been corrected to "CAPOX."
  • The comment regarding a higher mean BMI associated with survivors has been expanded. Additional references to similar studies have been included to support the "obesity paradox."
  • The authors' speculation on the NHL score as a potential prognostic marker has been elaborated. A brief explanation of the utility of biomarkers as prognostic factors has been added (lines 188-211) to enhance the scientific basis of this calculation.
  • The comment suggests including dihydropyrimidine dehydrogenase test data for the cohort if available. Unfortunately, such data are not available for inclusion in this study.
  • The discussion on the association between chemotherapy and observed biomarkers has been expanded. Additional information and a brief argument regarding their utility and potential applications have been included to provide a more detailed explanation (line 177-183).

We appreciate the insightful suggestions provided. They have been extremely useful in enhancing the clarity and comprehensiveness of our manuscript.

Reviewer 3 Report

Comments and Suggestions for Authors

Review Report  on the Influence of Biomarkers on Mortality Among Patients with He-2 patic Metastasis of Colorectal Cancer Treated with FOL-3 FOX/CAPOX and FOLFIRI/CAPIRI, Including Anti-EGFR and 4 Anti-VEGF Therapies

This  article is written with inadequate citations and inadequate references. In addition, author should need to define rationale to chose FOX/CAPOX and FOLFIRI/CAPIRI, and Anti-EGFR and 4 Anti-VEGF therapies over other existing therapies for CRC. Author should have to add some as a table of other existing biomarkers for CRC that has been developed and using in clinics and make comparison with biomarkers that they used  to test for their study. Author should have to outline mechanism of the therapies for CRC that they analyzed.

Sincerely

Author Response

Dear Reviewer,
Thank you for your detailed feedback on our article, "Influence of Biomarkers on Mortality Among Patients with Hepatic Metastasis of Colorectal Cancer Treated with FOLFOX/CAPOX and FOLFIRI/CAPIRI, Including Anti-EGFR and Anti-VEGF Therapies."
We appreciate your comments regarding the citations and references. We want to assure you that all references and citations have been formatted according to MDPI guidelines.
We acknowledge your suggestion to define the rationale for choosing FOLFOX/CAPOX and FOLFIRI/CAPIRI, along with Anti-EGFR and Anti-VEGF therapies over other existing therapies for CRC. We will address this in future articles to provide a more comprehensive comparison and rationale. Additionally, while we recognize the value of including a table of other existing biomarkers for CRC, unfortunately, we were unable to incorporate this information within the limited timeframe. However, we plan to include these biomarkers and make relevant comparisons in our upcoming publications.
Regarding the suggestion to outline the mechanisms of the therapies analyzed, we decided that including such detailed explanations would make the article excessively long. We aimed to maintain a focused discussion on the primary findings of our study.
Thank you again for your valuable feedback and for helping us improve our work.